# Delayed and Interrupted Ventilation with Excess Suctioning after Helping Babies Breathe with Congolese Birth Attendants

**DOI:** 10.3390/children10040652

**Published:** 2023-03-30

**Authors:** Jackie K. Patterson, Daniel Ishoso, Joar Eilevstjønn, Melissa Bauserman, Ingunn Haug, Pooja Iyer, Beena D. Kamath-Rayne, Adrien Lokangaka, Casey Lowman, Eric Mafuta, Helge Myklebust, Tracy Nolen, Janna Patterson, Antoinette Tshefu, Carl Bose, Sara Berkelhamer

**Affiliations:** 1Department of Pediatrics, University of North Carolina at Chapel Hill, 101 Manning Drive, CB 7596, Chapel Hill, NC 27514, USA; 2School of Public Health, University of Kinshasa, Kinshasa 11850, Democratic Republic of the Congo; 3Laerdal Medical, 4002 Stavanger, Norway; 4RTI International, Research Triangle Park, Durham, NC 27709, USA; 5American Academy of Pediatrics, Itasca, IL 60143, USA; 6Department of Pediatrics, University of Washington, Seattle, WA 98195, USA

**Keywords:** newborn resuscitation, Helping Babies Breathe, bag-mask ventilation

## Abstract

There is a substantial gap in our understanding of resuscitation practices following Helping Babies Breathe (HBB) training. We sought to address this gap through an analysis of observed resuscitations following HBB 2nd edition training in the Democratic Republic of the Congo. This is a secondary analysis of a clinical trial evaluating the effect of resuscitation training and electronic heart rate monitoring on stillbirths. We included in-born, liveborn neonates ≥28 weeks gestation whose resuscitation care was directly observed and documented. For the 2592 births observed, providers dried/stimulated before suctioning in 97% of cases and suctioned before ventilating in 100%. Only 19.7% of newborns not breathing well by 60 s (s) after birth ever received ventilation. Providers initiated ventilation at a median 347 s (>five minutes) after birth; no cases were initiated within the Golden Minute. During 81 resuscitations involving ventilation, stimulation and suction both delayed and interrupted ventilation with a median 132 s spent drying/stimulating and 98 s suctioning. This study demonstrates that HBB-trained providers followed the correct order of resuscitation steps. Providers frequently failed to initiate ventilation. When ventilation was initiated, it was delayed and interrupted by stimulation and suctioning. Innovative strategies targeting early and continuous ventilation are needed to maximize the impact of HBB.

## 1. Introduction

Each year, approximately 10 million newborns are born with respiratory depression, defined as failure to breathe at birth [1]. Respiratory depression may cause significant morbidity among survivors, and ultimately results in 900,000 neonatal deaths annually [2]. Most newborns who die from respiratory depression are born in low- and middle-income countries (LMICs) where the resources and trained staff to provide quality intrapartum and postnatal care are limited [3]. Many deaths from respiratory depression are preventable with basic resuscitation, including stimulation, clearing of the airway, and bag-mask ventilation (BMV). Of newborns with respiratory depression, approximately 40% will respond to stimulation alone and 50% to BMV [1]. Resuscitation training programs, such as Helping Babies Breathe (HBB), focus on these simple practices. HBB is a low-cost, simulation-based program designed for low-resource settings. For newborns who do not cry at birth, HBB recommends suctioning if needed, followed by stimulation. For newborns who do not breathe with stimulation, HBB recommends BMV initiated within the Golden Minute (i.e., by 60 s after birth). A meta-analysis of HBB studies published between 2010 and 2016 showed that HBB reduces 24 h newborn mortality by 30.7% [4].

Despite a decade of literature evaluating the impact of HBB, a gap remains in our understanding of the implementation of resuscitation steps following training. Most reports of resuscitation practices focus on timely BMV. A meta-analysis of four HBB studies demonstrated the rate of BMV within the Golden Minute increased by more than two and a half times after training [5]. Despite this increase, up to one third of BMV episodes were still initiated beyond the Golden Minute [5]. Two meta-analyses of HBB studies demonstrated no change in the frequency of use of stimulation, suction, or BMV [4,5]. Little is known about the initiation time, duration, and repetition of stimulation and suction and the duration and continuity of ventilation following training [6,7,8,9,10]. Additionally, previous studies have not provided details on the order in which providers implement resuscitation steps. Resuscitation practices that deviate from HBB may be less effective. A detailed understanding of these variations may assist educators in developing alternate strategies to maximize the impact of evidence-based resuscitation.

In a recent newborn resuscitation clinical trial in the Democratic Republic of the Congo (DRC), we gathered resuscitation data on provider practices and the newborn’s breathing status via direct observation [11]. This trial evaluated the impact of resuscitation training with continuous electronic heart rate (HR) monitoring on the primary outcome of total stillbirths. In this trial, providers received HBB training, regularly practiced their resuscitation skills in simulation, and incorporated a battery-operated HR meter called NeoBeat into their resuscitation care. Using the dataset from this trial, we sought to address the critical gap in understanding of resuscitation practices through a detailed analysis of care following HBB training. We hypothesized that providers initiate ventilation beyond the recommended Golden Minute and, despite de-emphasis of suctioning with HBB 2nd edition, suction frequently.

## 2. Materials and Methods

This is a secondary analysis of an interventional trial to evaluate the effect of resuscitation training with continuous electronic HR monitoring on identification of stillbirth [11,12]. We conducted this trial in three health facilities in Kinshasa, DRC (referred to as facilities A, B, and C) with delivery census ranging from 1051 to 4248 annual births. In this manuscript, we report direct observations of resuscitation care during the period following HBB training.

The trial included all in-born neonates with the exclusion of newborns <28 weeks gestation (or birthweight < 1000 g if gestational age unavailable). For this secondary analysis, we included liveborn neonates (as classified in the medical record) born via non-instrumented vaginal delivery between 10/25/18 and 7/28/19 whose resuscitation care was observed by research staff. We excluded fresh stillbirths from this analysis due to concern that provider resuscitation practices may be influenced by this diagnosis.

Providers of newborn care were primarily midwives but included physicians for complex resuscitations at facility B where cesarean sections were also offered. The education of the midwives ranged from lower secondary school (~17%) to a bachelor’s degree (~40%) to a master’s degree with licensing in midwifery (~43%). Some midwives had prior exposure to HBB. Depending on the rate of annual births at their facility of employment, midwives typically attended 58–233 births annually. Physicians infrequently attended births. All facilities had a functioning bag and mask that were used during resuscitations prior to initiation of the interventional trial; we also provided an upright bag and mask with the NeoNatalie manikin kit, which providers chose to adopt clinically. Facilities A and C had a wall-mounted clock with a second hand in the delivery area that could be used by providers to keep track of time during resuscitation.

Experienced, in-country HBB master trainers conducted one-day in-service HBB workshops for all newborn resuscitation providers at each facility in August 2018 (approximately two months prior to the start of data collection). The master trainers used the standard HBB 2nd edition training materials in French (the primary language of the providers). Participants practiced skills in pairs using the NeoNatalie manikin, upright bag and mask, and penguin suction device. For the assessment of the participants, the study team adapted the HBB knowledge check to include a question about evaluation of HR to determine vital status. We also adapted the objective structured clinical exam (OSCE) A to include the placement of an HR meter. Before and after training, participants completed this knowledge check, the HBB BMV skills checklist, and OSCE A.

Following training, HBB action plans were posted in the delivery ward at each facility. Participants received a personal copy of the provider guide for individualized review. A NeoNatalie manikin and resuscitation equipment were provided to each facility to support regular practice of skills. Master trainers recommended simulation practice to be performed once weekly or as practical. Providers practiced skills throughout the trial by engaging in OSCEs and received verbal feedback from the medical director and/or head nurse midwife.

We also trained providers in the use of a battery-operated HR meter called NeoBeat^TM^ (Developer: Laerdal Global Health, Stavanger, Norway) [13]. NeoBeat uses dry-electrode technology incorporated into a C-shaped device and can be placed on a newborn within 3 s by a single provider [14]. In all facilities, providers placed NeoBeat on non-breathing newborns and then continued with resuscitation. Facility A participated in a sub-study evaluating HR in breathing newborns; at this facility, midwives placed NeoBeat on all newborns at birth.

Six research staff with a clinical background (either nursing or physician training) collected data for this study. All research staff underwent HBB training to ensure familiarity with resuscitation practices for the purposes of data collection. Research staff collected demographic data from each facility’s delivery register and directly observed a convenience sample of resuscitations. At facility A, staff observed all births for 160 h weekly; at facilities B and C, staff were present for 40 h weekly and only observed newborns that did not breathe at birth [11]. Using a tablet application called Liveborn (co-developed by Laerdal Medical and the University of North Carolina at Chapel Hill), research staff documented the start and stop time of resuscitative practices (drying/stimulation, suction, BMV, skin-to-skin), the time of cord clamping, and the timing and occurrence of corrective steps to improve BMV (Figure 1) [15]. They also documented the breathing status of the newborn continuously using the following categories: not breathing, gasping/shallow breathing, or breathing well.

To ensure an accurate data collection during observation, we developed a training manual with detailed instructions for recording observations that outlined clear definitions for each variable in the Liveborn app, including resuscitation care and the breathing status of the newborn. Following training, research staff practiced using the app by observing and documenting 10 video-taped clinical resuscitations. Three physicians (CB, DI, and JKP) developed a gold standard annotation for the 10 videos used in the training. We considered research staff observations sufficiently accurate if they included all resuscitative practices and documented a start and stop time of each practice within +/− 3 s of the gold standard. Once staff achieved sufficient accuracy using Liveborn for video-taped resuscitations, they practiced observing resuscitations in the clinical environment for one month with intermittent evaluation of their proficiency by DI. All research staff achieved competency in observation prior to the start of data collection.

We defined key resuscitation actions per Appendix A. Each event registered with a start and stop time was considered an ‘episode’ with its duration corresponding to the number of seconds elapsing between that start and stop time. We defined breathing well as breathing regularly (~40 breaths per minute) or crying per the description in HBB [5]. We categorized newborns into those breathing well at birth versus not breathing well at birth, defined by their breathing status at 30 s after birth. We defined delayed cord clamping as clamping the cord after 60 s. Of note, while suctioning is a step in improving ventilation, all instances of suctioning were logged and analyzed as suctioning rather than improving ventilation.

We used descriptive statistics to evaluate the breathing status of the newborn over time, along with the frequency, timing of initiation, duration, and order of key resuscitation practices. We used means and standard deviations for normally distributed data, and medians and quartiles for non-normal data distributions.

The University of North Carolina Institutional Review Board (IRB) and the local DRC National Ethics Committee IRB approved the interventional trial (UNC IRB #17-1688; DRC IRB #058/CNES/BN/PMMF/2017). A full waiver of informed consent was granted for this study. Due to concurrent participation in a study of electronic HR monitoring in well babies in facility A that required informed consent and the inability to predict which babies would not breathe at birth, informed consent was obtained for all participants at that site. Informed consent was standardly obtained from the newborn’s guardian prior to birth; however, when consent was unable to be obtained prior to birth, observational data were collected, and consent was obtained after birth before storing the data.

## 3. Results

### 3.1. Knowledge and Skills after HBB Training

Providers (*n* = 59 [45 midwives, 14 physicians]) performed at a median score of 79% on the post-knowledge check (Table 1). The median scores for the BMV skills check were 36% and 79% pre- and post-training, respectively. The median OSCE A scores were 43% and 79% pre- and post-training, respectively. On average, providers practiced their skills with OSCEs once monthly throughout the nine-month period of HBB implementation.

### 3.2. Newborn Demographics

Among the 2592 liveborn neonates for whom we captured observational data, 9.3% were low birthweight (<2500 g) and 10.8% were premature (Table 2). 98.2% of newborns received care at facility A. While newborns from facilities B and C made up only 1.7% of this cohort, all of these newborns were not breathing well by 30 s after birth (a reflection of the difference in the observational strategy at facility A).

### 3.3. Newborn Respiratory Status

Among newborns at facility A (where we gathered observational data on both breathing and non-breathing newborns), we observed a steep decline in the percentage of non-breathing newborns between 30 s and 60 s after birth, followed by a change in the slope from approximately 90 s onwards (Figure 2). Among newborns at facility A, 28.1% were not breathing well at 30 s and 12.1% were not breathing well at 60 s after birth. At Facility A, the rate of BMV was 2.0%. Among all newborns in the trial not breathing well at 60 s, 19.7% eventually received BMV.

### 3.4. Order of Resuscitation Practices

Among newborns breathing well, providers dried 31.1% prior to placing skin-to-skin, and 99.5% received skin-to-skin care prior to cord clamping (Appendix A). Among newborns not breathing well, drying/stimulation preceded suctioning in 97.0% of cases. Drying/stimulation, suctioning, and cord clamping each preceded ventilation in 100% of cases.

### 3.5. Timing of Resuscitation Practices

Among newborns breathing well, 89.9% received skin-to-skin at a median 6 s after birth (Table 3). All newborns received drying/stimulation at 10–11 s after birth. Providers spent a median of 42 s drying/stimulating newborns breathing well, 62 s drying/stimulating newborns not breathing well, and 132 s (>2 min) drying/stimulating ventilated newborns. Providers clamped the cord at 124 s after birth for newborns breathing well and 159 s for newborns not breathing well.

Providers suctioned 56.7% of newborns breathing well, 78.6% of newborns not breathing well, and 100% of ventilated newborns (Table 3). Suctioning was initiated beyond the Golden Minute with a median start of 76 s from birth for ventilated newborns, 90 s for newborns not breathing well, and 138 s for newborns breathing well. Providers spent a median of 50 s suctioning babies breathing well, 68 s suctioning babies not breathing well, and 98 s suctioning ventilated newborns. The average duration of one suctioning episode ranged from a median 30 s among ventilated newborns to 38 s among newborns breathing well.

When providers initiated ventilation, it was initiated beyond the Golden Minute with a median of 326 s (>five minutes) after birth for newborns not breathing well at 30 s after birth (Table 3). No episodes of BMV were initiated in the Golden Minute. Both drying/stimulation and suctioning contributed to delays in BMV (Figure 3). An amount of 72% of initial BMV episodes lasted <60 s. On average, pauses between episodes of BMV lasted a median 22 s. During pauses in BMV, stimulation and suctioning were commonly performed (Figure 3). Among resuscitations involving BMV, providers took corrective steps in 72% of cases.

## 4. Discussion

After HBB training, providers followed the correct order of resuscitation steps for babies not breathing well at birth. Based on a recommendation for use of BMV within the Golden Minute, providers frequently failed to initiate ventilation for non-breathing infants. When providers performed BMV, it was both substantially delayed and frequently interrupted by stimulation and suctioning.

Providers were generally adherent to actions recommended for every birth, such as drying, initiation of skin-to-skin, and delayed cord clamping. These practices are critical for avoiding hypothermia, supporting early initiation of breastfeeding, and optimizing blood volume and iron stores [16]. Our findings support prior studies that demonstrate compliance with drying, [6] initiation of skin-to-skin, and delayed cord clamping following HBB [4]. The opportunities to practice these actions are frequent, and the circumstances in which they are practiced are often low-stress, as newborns who successfully complete the cardiorespiratory transition at birth still require this care. We postulate that frequent practice of these common steps with babies who are breathing well and the simplicity of these actions enable providers to consistently perform these steps for newborns who require resuscitation.

Providers overused suctioning, both by suctioning newborns who cry at birth and by repeating and prolonging suctioning for newborns who fail to breathe. This preoccupation with suctioning is concerning in light of literature substantiating the adverse effects of this practice. In a summary of the evidence, the International Liaison Committee on Resuscitation stated that evidence supporting oropharyngeal/nasopharyngeal suctioning is limited, and the practice can have serious side effects including vagal-induced bradycardia [17], increased risk of infection [18,19], lower oxygen saturation [19,20,21,22], apnea [23], neonatal brain injury [24,25], and other side effects [26,27,28,29]. Of particular concern is the time required to suction, which can delay the initiation of life-saving BMV [20,30]. While several evaluations of HBB have demonstrated a decrease in suctioning following training [6,8], two meta-analyses demonstrated no change in the frequency of suctioning [4,5]. In light of the concerns for excess use of suctioning, HBB 2nd Edition addressed and revised the indications for suctioning. Specifically, the updated curriculum eliminated the use of suctioning for meconium prior to drying and changed the indications for suctioning to be performed only if the airway was obstructed or if there was meconium in the amniotic fluid and the baby was not crying after drying. Despite training in this updated curriculum, providers continued to overuse suctioning in our study [31,32]. Given the over-application of suction to newborns who do not need it (especially crying newborns), we postulate that providers become very comfortable with this practice and overestimate its importance in resuscitation. As such, providers may be quick to resort to and return to suctioning when faced with the stress of resuscitating a non-breathing newborn.

Based on the recommendation to initiate BMV within the Golden Minute for non-breathing newborns, providers frequently failed to initiate BMV as part of their resuscitation care. In our cohort from facility A, 12.1% of liveborn infants were not breathing well by one minute, consistent with a prior multi-country study in LMICs [33]. Only 19.7% of newborns who were not breathing well by one minute ever received BMV in our trial. While most of these newborns ultimately established effective, spontaneous respirations without BMV, their cardiorespiratory transition was delayed, and the long-term impact of this delay remains a concern. The 2.0% rate of BMV in this study is low compared to prior studies of HBB training that have demonstrated rates of 3.0–11.7% following HBB training [5,8,9].

Of additional concern is the substantial delay in initiation of BMV observed among HBB-trained providers. Timely establishment of ventilation is critical to reducing morbidity and mortality associated with perinatal respiratory depression. For every 30 s delay in BMV, the risk of death increases by 16% [30,34]. The delay in the initiation of BMV reported in this study is markedly different from prior studies of HBB demonstrating up to three quarters of BMV episodes are initiated within the Golden Minute [4]. We speculate that the difference in these results may partially reflect our pragmatic implementation of HBB that included practice of BMV skills only once monthly. Our results are more consistent with practices of HBB-trained providers distal to training, such as a recent study in Bangladesh, Nepal, and Tanzania, which demonstrated < 1% of BMV was initiated within one minute of birth [33]. The time to BMV in our cohort is similar to a prior study of resuscitation practices among midwives in Mozambique that demonstrated a 220 s median time to BMV [35]. Based on previously published focus group discussions with midwives from this trial, it is unlikely that NeoBeat contributed to the delay initiating BMV, as providers did not perceive that NeoBeat distracted them from their resuscitation care [36]. Given standard practice in these facilities of cutting the cord prior to ventilation, the 164 s median time of cord clamping among ventilated cases suggests that delayed cord clamping may have critically delayed the initiation of BMV. Additionally, repeated and prolonged stimulation and suctioning contributed to delayed BMV in this study. This is also consistent with prior literature demonstrating HBB-trained Tanzanian midwives spent an average of 23 s stimulating and 22 s suctioning prior to initiating BMV [10]. The substantial time spent stimulating and suctioning non-breathing newborns suggests the delay in BMV is not entirely attributable to delayed recognition that a newborn needs help to breathe. We postulate that when providers recognize a need for resuscitation, they hesitate to initiate the less practiced and more complex skill of BMV. Instead, providers rely on routine actions they are more comfortable executing, such as stimulation and suctioning.

In addition to delayed BMV, we observed frequent pauses in BMV. Continuous ventilation is critical; interrupted BMV increases the risk of death by 75% [37]. Our findings are similar to a study of HBB-trained Tanzanian midwives, which showed pauses between the first two episodes of BMV lasted a median 8 s [10]. In our cohort, many pauses between episodes of BMV were spent repeating the actions of stimulation or suctioning. We postulate providers may abandon BMV when the newborn does not respond quickly, returning to routine actions they are more comfortable executing in hopes they will revive the newborn.

The strengths of this study include our detailed data collection with direct observation of provider actions and the newborn’s respiratory status over time, all captured in a large sample of births. However, we also acknowledge limitations. We are unable to compare resuscitation practices following HBB to practices before training, so we cannot comment on changes in care with HBB. Similarly, we cannot determine whether HR display from NeoBeat influenced resuscitation practice in addition to HBB training. We did not collect data on which provider was resuscitating and, thus, cannot comment on individual clinical performance. We did not collect detailed data on monthly simulation practice. Although HBB 2nd edition was used, staff did not receive any external support to engage in HBB-related QI efforts following training, and we did not document any of these efforts. While research staff documenting resuscitation actions on Liveborn underwent rigorous training to support accurate data collection, we acknowledge potential inaccuracy and limitations compared to annotating videos. We used a subjective evaluation of the newborn’s breathing status as determined by research staff rather than objectively evaluating respiratory status using an electronic monitor. We did not evaluate the effectiveness of BMV nor the effectiveness of steps to improve ventilation.

Our findings reflect a pragmatic approach to HBB with one-day training workshops followed by monthly practice of skills. The resuscitation practices observed in this study may reflect deficiencies in the initial training or inadequacies in the subsequent LDHF practice. However, similar variance from HBB recommendations has been reported in the context of rigorous LDHF training in Tanzania [10]. Our findings underline the importance of investment in on-going LDHF practice as a critical piece of HBB implementation. Furthermore, we speculate that strategies beyond training, such as quality improvement, expert guidance, directed mentoring, and debriefing, may be necessary to improve adherence.

## 5. Conclusions

HBB-trained providers followed recommendations for routine care and performed resuscitation steps in the correct order for non-breathing newborns. However, providers overused suction (suctioning frequently and for prolonged duration) and failed to initiate BMV. When providers initiated BMV, it was both substantially delayed and interrupted by repetitive stimulation and suctioning. Innovative strategies targeting early and continuous ventilation are needed to optimize the impact of HBB.

## Figures and Tables

**Figure 1 children-10-00652-f001:**
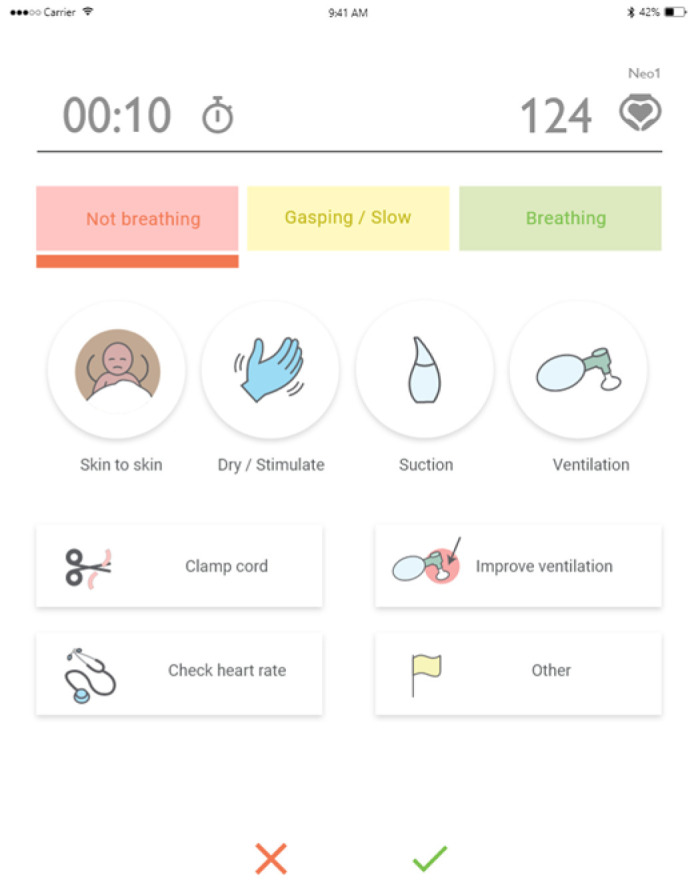
Event registration screen of the Liveborn app used for observational data collection.

**Figure 2 children-10-00652-f002:**
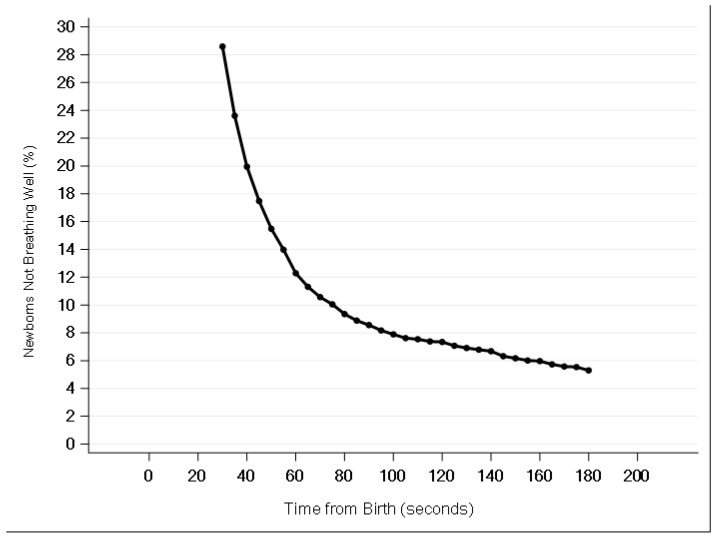
Liveborn neonates not breathing well at 30 s after birth from Facility A: breathing status from 30 s to 3 min after birth.

**Figure 3 children-10-00652-f003:**
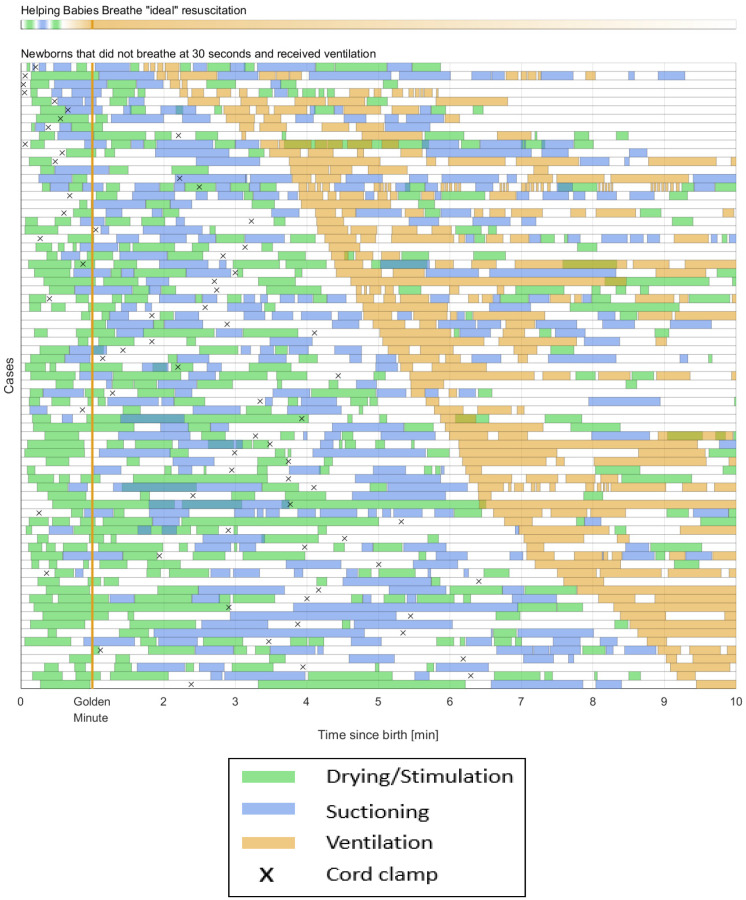
Resuscitation care practices among newborns not breathing well by 30 s after birth who received bag-mask ventilation (BMV; *n* = 73). The Helping Babies Breathe (HBB) “ideal” resuscitation demonstrates the practices recommended by HBB for a non-breathing baby, starting with drying shortly after birth, followed by brief suctioning (if secretions are noted), and returning to stimulation before initiating BMV by one minute after birth. Below, each row represents an individual newborn who received BMV.

**Table 1 children-10-00652-t001:** Provider knowledge and skill.

Assessment % Correct	Participant Scores *n* = 59 Median (Quartiles)
Knowledge check (out of 19 items)	
Before	68 (58, 82)
After	79 (74, 87)
BMV ^1^ skills check (out of 14 items)	
Before	36 (21, 57)
After	79 (68, 93)
OSCE ^2^ A (out of 14 items)	
Before	43 (29, 57)
After	79 (71, 86)

^1^ BMV = bag-mask ventilation. ^2^ OSCE = objective structured clinical exam.

**Table 2 children-10-00652-t002:** Demographic characteristics of livebirths.

Characteristic	Livebirths *n* = 2592 *n* (%)
Maternal age ^1^	
<20 20–35 >35	163 (6.3) 2094 (80.9) 332 (12.8)
Parity ^2^ 0 1–2 ≥3	
	825 (32.0)
	1057 (40.9)
	700 (27.1)
Birthweight 1000–1499 g 1500–2499 g ≥2500 g	
	9 (0.3)
	233 (9.0)
	2350 (90.7)
Gestational age ^3^ <37 weeks ≥37 weeks	
	260 (10.8)
	2149 (89.2)
Small for gestational age ^3^ Yes No	
	196 (8.1)
	2213 (91.9)
Multiplicity Singleton Twins	
	2501 (96.5)
	91 (3.5)
Newborn Sex ^1^ Male Female	
	1321 (51.0)
	1268 (49.0)
Facility of Birth ^4^ A B C	
	2546 (98.2)
	14 (0.5)
	32 (1.2)

^1^ *n* = 3 missing (0.1%); ^2^ *n* = 10 missing (0.4%); ^3^ Gestational age defined by last menstrual period; *n* = 183 missing (7.1%); ^4^ Livebirths at facilities B and C were only eligible for observation if they did not breathe at birth; 29 of the 46 newborns at these two facilities received bag-mask ventilation.

**Table 3 children-10-00652-t003:** Timing and duration of resuscitation care practices for liveborn neonates.

Resuscitation Care Practice, Median (Quartiles) ^1^	Breathing Well by 30 s after Birth *n* = 1818	Not Breathing Well by 30 s after Birth *n* = 774	Ventilated ^2^ *n* = 81
Skin-to-Skin, *n* (%) Initiation time from birth	1635 (89.9) 6 (3, 14)	681 (88.0) 7 (3, 16)	51 (63) 11 (5, 23)
Drying/Stimulation, *n* (%)	1816 (99.9)	774 (100.0)	81 (100)
Number of episodes per newborn	3.0 (2.0, 3.0)	3.0 (2.0, 5.0)	6.0 (4.0, 8.0)
Initiation time from birth	10 (6, 17)	11 (7, 17)	11 (7, 22)
Duration of first episode	17 (11, 29)	20 (13, 32)	17 (11, 29)
Total duration of all episodes	42 (29, 59)	62 (44, 94)	132 (82, 177)
Average duration of each episode	16 (11, 22)	19 (14, 27)	19 (15, 27)
Average time between each episode	95 (61, 151)	79 (51, 121)	76 (53, 106)
Cord Clamp, *n* (%) Initiation time from birth	1785 (98.2) 124 (95, 164)	751 (97.0) 159 (122, 209)	77 (95) 164 (63, 226)
Suction, *n* (%)	1030 (56.7)	608 (78.6)	81 (100)
Number of episodes per newborn	1.0 (1.0, 2.0)	2.0 (1.0, 3.0)	3.0 (2.0, 6.0)
Initiation time from birth	138 (70, 250)	90 (54, 205)	76 (45, 146)
Duration of first episode	39 (26, 56)	38 (24, 57)	34 (20, 51)
Total duration of all episodes	50 (32, 73)	68 (43, 108)	98 (62, 171)
Average duration of each episode	38 (26, 54)	36 (25, 55)	30 (20, 42)
Average time between each episode	90 (34, 145)	79 (34, 127)	64 (34, 87)
Bag-Mask Ventilation, *n* (%)	8 (0.4)	73 (9.4)	81 (100)
Number of episodes per newborn	2.0 (1.0, 3.0)	3.0 (1.0, 5.0)	3.0 (1.0, 5.0)
Initiation time from birth	437 (380–583)	326 (103–567)	347 (103–583)
BMV initiated prior to 60 s, *n* (%)	0 (0)	0 (0)	0 (0)
First episode lasting ≥ 60 s, *n* (%)	5 (62)	18 (25)	23 (28)
Duration of first episode	81 (49, 122)	26 (16, 60)	30 (17, 71)
Total duration of all episodes	144 (84, 206)	83 (55, 173)	93 (58, 189)
Average duration of each episode	71 (40, 137)	28 (18, 47)	31 (19, 58)
Average time between each episode	8 (8, 18)	23 (13, 44)	22 (12, 40)
Improve Ventilation, *n* (%) ^3^	3 (37.5)	55 (75.3)	58 (72)
Number of episodes	1.0 (1.0, 6.0)	2.0 (1.0, 4.0)	2.0 (1.0, 4.0)
Initiation time from birth	490 (457, 548)	366 (310, 460)	380 (313, 473)
Average time between each episode	30 (30, 30)	28 (0, 59)	30 (0, 59)

BMV = bag-mask ventilation ^1^ All time is reported in seconds ^2^ Ventilated neonates represent the subset of liveborn neonates who received BMV regardless of breathing status at 30 s ^3^ Improve ventilation episodes include reapplying the mask, repositioning the head, opening the mouth slightly or squeezing the bag harder. Percentage denominator = all resuscitation cases receiving BMV.

## Data Availability

The data presented in this study will be openly available in the USAID Data Development Library at data.usaid.gov, accessed on 10 February 2023.

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
