# Peer review of "Delayed and Interrupted Ventilation with Excess Suctioning after Helping Babies Breathe with Congolese Birth Attendants"

_children, 2023, doi:10.3390/children10040652_

Round 1
Reviewer 1 Report
Dear editor,
Thank you for the opportunity to review the manuscript. This secondary analysis of the NeoBeat trial is well written and presents valuable data for the readers interested in the STG goal of reducing neonatal death. Data registration by the resuscitation table uses an innovative method. The number of resuscitation events and detail of descriptions is impressive. However, the presentation of the result section could be confusing to many reader. Data was recorded differently at facility A compared to facilities B and C. Yet results are presented as agglomerated data in the same tables. It surprises me that only 1,7% of the data comes from facilities B and C even if only resuscitation practice was recorded. Was there more missing data or was there a big difference in volume of deliveries? Please explain this and motivate the reason for including facilities B and C. Overall, an original study presenting important problems that will need to be improved if STG goals are to be met.
Specific comments
Line 75-76
Additional information regarding the setting such as number of deliveries in facilities A,B and C would be useful.
Line 89-90
Authors state that all facilities had functional bag and masks prior to the trial. Experience from our research team across a wide range of facilities across the continent is that this is not the case, with many ill-fitting, leaking devices. Many HBB interventions include providing functional equipment to facilities where training occurs. Inefficient ventilation could oblige health providers to alternate to other resuscitation methods. Please confirm that NeoBeat trial did not provide new equipment.
Line 114-115
How many research assistants participated in the study? Did they have a clinical background? Was data collected 24/7 during the 9 month period in all three facilities?
Line 185
Could you clarify the BMV rate in relation to the total number of deliveries?
Line 191
99,5% of newborns received skin to skin care prior to chord clamping. Was this only for vaginaly delivered babies? It would be easier to understand this number if C-section rate at facility B was reported.
Supplemental Table 2
Please clarify what you mean by resuscitation including both practices
Table 3
Clarification is needed for the numbers on each line such as (s) for seconds and what the values are in the parenthesis.
The last part of the table ’improve ventilation’ is hard to understand.
Figure 3 provides an interesting visual representation of the resuscitation events. However, the number of events seem to exceed by far those reported in table 3. Please explain!
It would have been useful to also get a visual cue for when the chord was clamped for each case.
Line 238
Please rephrase and avoid euphemism ’underused ventilation’ when no ventilation was provided.
Line 245
Since ventilation on intact chord barely was practiced, the data suggests that the average 164 seconds of delayed chord clamping could have critically delayed initiation of ventilation
Line 269
Replace over-inflate by overestimate
Line 275
If I read the results correctly, only 3,5% of newborn received BMV. This number can also be reported and contrasted with reports of similar settings in Tanzania and Uganda, where 8-10% of all newborn received BMV.
Line 300
This cohort differs from ordinary low-resource setting, because HR data was visible to the providers because of the NeoBeat . Did the HR display on the newborn influence practice?
Reviewer 2 Report
Very interesting manuscript by Patternson et al about HBB training and outcomes in Congo.
Some of these comments author may want to address to increase clarity of manuscript without referring to their other manuscripts
1. Would it be possible for authors to tell briefly about the clinical trial of resuscitation training with continuous electronic heart rate monitoring along with what primary outcome was in the introduction section?
2. Did providers receive any additional training during the trial or they only practiced HBB skills on their own?
3. Were research staff observing had HBB training? What were their qualifications?
4. What was the time frame between HBB training and start of the research study?
5. Were there any clocks in delivery room to prompt providers to achieve certain steps?
6. Was “not breathing well” clearly defined to the providers before start of the study?
7. Did the providers receive any feedback on how they performed the HBB steps after certain number of deliveries they attended?
8. Was there any plan to have interim data looked at by investigators and give feedback to providers about their performance?
9. Authors may want to add in the discussion what steps can we take to improve performance of the providers to make it more successful transition for the babies
10. Is there data on outcome of the babies who received delayed PPV?
11. Was there any correlation between the providers who did not perform well in the testing and the resuscitation they provided at the time of delivery?
